# Lived Experience of Men with Prostate Cancer in Ireland: A Qualitative Descriptive Study

**DOI:** 10.3390/healthcare13091049

**Published:** 2025-05-02

**Authors:** Seidu Mumuni, Claire O’Donnell, Owen Doody

**Affiliations:** School of Nursing and Midwifery, University of Limerick, V94 T9PX Limerick, Ireland; seidu.mumuni@ul.ie (S.M.); claire.odonnell@ul.ie (C.O.)

**Keywords:** prostate cancer, persons, experiences, men

## Abstract

Background: Prostate cancer is recognised as the second most common diagnosed cancer in men and remains a significant global public health concern. In Ireland, the incidence of prostate cancer continues to rise, with approximately 1 in 6 men being diagnosed in their lifetime. Men’s experiences with prostate cancer are complex, necessitating further research into the factors influencing diagnosis and treatment. Therefore, this study aims to explore men’s experiences with prostate cancer, emphasising the interplay between screening, diagnosis, and the lived experiences of those affected. Methods: A qualitative descriptive study was conducted among men with prostate cancer in Ireland. Using a purposive sampling (n = 11) were interviewed with data saturation guiding sample size determination. A semi-structured interview guide was used for data collection either face-to-face or via Microsoft Teams and phone calls. Data were analysed using Braune and Clarke’s thematic analysis approach after transcription, with NVivo 12.0 software supporting analysis. Results: Thematic analysis identified five themes: systemic obstacle in timely cancer detection, the role of efficient system in cancer care, emotional resilience in cancer recovery, redefining normalcy post treatment and harnessing specialised support network in coping strategies. These themes were examined through the lens of the Biopsychosocial Model to understand their interconnected nature and impact on patient experiences. Conclusions: This study highlights the complex factors affecting prostate cancer patients’ experiences, emphasizing the need for a patient-centred approach, addressing systemic disparities, and promoting multidisciplinary care. It suggests implementing evidence-based survivorship care frameworks to improve quality of life for survivors, with future research exploring long-term effects of integrated care models.

## 1. Introduction

Prostate cancer, the second most prevalent malignancy among men globally, remains a significant health concern with complex implications for those affected [1,2]. With a staggering 1,467,854 new cases and over 397,430 deaths in 2022 alone [3], this disease is a profound public health issue. According to the Irish Cancer Society (ICS) over 4000 men are diagnosed with prostate cancer annually in Ireland, meaning approximately one in six men will be diagnosed in their lifetime [4]. The disease predominantly originates in the peripheral zone of the prostate (70% of cases), with central and transition zone cancers comprising 10% and 20%, respectively [4]. Although adenocarcinoma accounts for 95% of prostate cancers, rarer forms including primary carcinoid, sarcoma, and primary small-cell carcinomas, highlighting the need for targeted screening and early detection strategies [5].

Prostate cancer incidence varies significantly across regions, with rates in Oceania and North America being over six times higher than in Asia, suggesting an interplay of genetic, environmental, and lifestyle factors [2,6,7]. Age is a major risk factor, with incidence rising from 1 in 350 men under 50 years to 1 in 52 by ages 50–59, and further escalating in those over 65, who account for 55% of prostate cancer deaths [1,8]. Despite high survival rates, advanced prostate cancer frequently metastasizes to the bone, lymph nodes, lungs, and liver, leading to severe complications that affect both quality of life and survival outcomes [9]. These challenges highlight the need for comprehensive screening programs, such as prostate-specific antigen (PSA) testing, despite ongoing debates about its potential for overdiagnosis [10].

Focusing on Ireland is particularly important, as prostate cancer remains one of the most diagnosed malignancies among Irish men. While advancements in early detection and treatment have improved outcomes, the Irish healthcare system faces challenges related to screening accessibility, specialist availability, and psychosocial support. Issues such as long waiting lists, limited access to urologists, and inconsistencies in screening recommendations contribute to diagnostic delays, which can negatively impact survival and quality of life. Furthermore, cultural and societal factors may influence men’s health-seeking behaviours, awareness levels, and willingness to undergo screening. Addressing these gaps in knowledge is crucial to understanding the barriers men face and informing policy decisions aimed at improving prostate cancer care in Ireland.

Current evidence suggests that modifiable lifestyle factors, including diet, smoking, and alcohol consumption, contribute to prostate cancer risk alongside non-modifiable factors such as age, race, and family history [11,12]. African American and Caribbean men face up to three times the risk compared to Caucasian men, while individuals with a family history of prostate cancer are at significantly increased risk, with the likelihood escalating based on the number of affected relatives [12,13,14]. As a result, many health organizations, including the US Preventive Services Task Force [15], recommend PSA testing for men aged 55–69, though there remains ongoing debate regarding mass screening [16].

The European Association of Urology opposes universal screening, instead advocating for early detection among high-risk men [16]. Given this complex landscape, research into men’s experiences with prostate cancer screening and diagnosis is essential, not only to address clinical challenges but also to explore how these factors impact men’s lives. In Ireland, where many cases are diagnosed incidentally or at later stages due to insufficient proactive screening, understanding patient experiences can bridge gaps in early detection and intervention strategies. Beyond survival, prostate cancer presents men with substantial physical and psychological challenges. Urinary difficulties, including weak flow, bladder fullness, painful urination, and incontinence, are common, alongside back pain and nocturia [17,18,19,20]. The indolent nature of prostate cancer means symptoms often develop silently, leading to potentially curable cases going undetected [1,11]. This reinforces the importance of understanding Irish men’s perspectives on screening, treatment decisions, and post-treatment support. Given Ireland’s healthcare infrastructure and cultural dynamics, a deeper examination of patient experiences can help identify gaps in support services and improve prostate cancer care [6,21].

Psychologically, a prostate cancer diagnosis often elicits significant distress, manifesting as anxiety, depression, and diminished self-esteem [6,22]. Many men experience concerns over treatment-related side effects, particularly those affecting masculinity, sexuality, and bodily functions, which contribute to emotional and psychological burdens [23,24]. Erectile dysfunction and urinary incontinence, for instance, can undermine self-worth and strain relationships with partners and families [25,26]. These concerns highlight the importance of research exploring Irish men’s specific experiences, including their emotional responses to diagnosis, decision-making processes, and available support structures. Insights from this research could inform healthcare policies that prioritize not only medical treatment, but also psychosocial interventions tailored to men’s unique needs.

Prostate cancer disproportionately affects racial and sexual minority groups, who may experience stigma, healthcare disparities, and lower screening uptake rates. African American men, for example, often report negative healthcare experiences, deterring them from timely screening [27]. Similarly, LGBTQ+ individuals face barriers to screening, though some studies suggest that gay and bisexual men may be more proactive in seeking screening options [28,29]. These disparities underscore the need for culturally sensitive, inclusive research to fully understand the diverse experiences of men with prostate cancer.

Addressing these challenges requires interventions that improve communication, accessibility, and reduce societal stigma. Given the complexity of men’s experiences with prostate cancer, further research is needed to examine factors influencing both diagnosis and treatment. By focusing on Ireland, this study aims to generate insights that will enhance screening initiatives, improve healthcare delivery, and inform policies that recognize the diverse ways prostate cancer impacts men’s lives. Ultimately, understanding these experiences will contribute to better health outcomes, increased awareness, and a more patient-centred approach to prostate cancer care in Ireland.

## 2. Methodology

### 2.1. Study Design and Settings

The study employed a qualitative descriptive design to explore the lived experiences of men with prostate cancer in Ireland. This design was chosen as it allows for an in-depth exploration of participants’ experiences, enhancing the depth and authenticity of the study [30]. To facilitate participant recruitment, the research team obtained ethical approval and permission to engage with the Irish Cancer Society and other relevant prostate cancer and men’s health organizations, such as Men’s Sheds. These organizations provide crucial support networks, offering Irish men a space to address various health and social concerns impacting both their communities and individual well-being.

### 2.2. Study Population and Eligibility Criteria

The study included men living with prostate cancer in Ireland, recruiting participants through various men’s health and prostate cancer organizations. To capture a diverse range of experiences, the study included men who were recently diagnosed, undergoing treatment, or in post-treatment stages. However, individuals who were critically ill or had cognitive impairments were excluded.

### 2.3. Sample Size and Sampling Procedure

The study employed purposive sampling to recruit participants. Information about the study, including an invitation letter, email, and participant information leaflet, was disseminated through relevant organizations. Additionally, study promotion was conducted via leaflet distribution and radio announcements. A total of 11 men participated, with the sample size determined based on the saturation principles (theoretical, data, code/theme and meaning) outlined by Rahimi [31].

### 2.4. Data Collection Tool

This study utilized a semi-structured interview guide, developed based on a review of existing literature and adapted to align with the study’s objectives. In-depth data on the lived experiences of men with prostate cancer were collected through face-to-face, Microsoft Teams, or phone interviews. Pretesting was conducted to ensure the relevance and effectiveness of the interview questions in the data collection process.

### 2.5. Data Collection Procedure

The interviews were conducted by the lead investigators (SM, OD, and CO’D) between March 2024 and February 2025. Depending on participant preference, interviews were held face-to-face, virtually via Microsoft Teams, or by telephone. Probing questions were used to explore responses in greater depth and clarify participants’ perspectives. Field notes were taken to capture non-verbal cues and contextual details that could not be recorded by the audio device. The in-depth interviews lasted between 30 and 40 min.

### 2.6. Data Analysis

All interviews were audio-recorded and transcribed verbatim. The transcribed data were uploaded to NVivo 12.0 software(QSR International, Melbourn, Australia) for data organization and text coding. The principal investigators independently conducted iterative readings of the transcripts with an open, exploratory approach to gain a comprehensive understanding of each participant’s narrative. Coding was employed to identify patterns within the text, aligning with the predetermined categories outlined in the research objectives. After multiple transcript reviews [32], thematic analysis framework (see Table 1) was applied to facilitate theme development and deeper interpretation. Categories were constructed based on key phrases within text segments, and sub-themes were identified within each category by the lead investigators.

To ensure credibility and alignment with the study’s objectives, the authors examined emerging meanings from the analytic outputs, categories, and themes (see Figure 1).

### 2.7. Ethical Considerations

As a registered nurse, the researcher adhered to the Code of Professional Conduct and Ethics for Registered Nurses and Registered Midwives [33]. Ethical approval for this study was granted by the Faculty of Education and Health Sciences Research Ethics Committee at the researcher’s university on 13 March 2024 (Reference No: 2023_12_10_EHS). All participants provided written informed consent before data collection for face-to-face interviews. For interviews conducted via Microsoft Teams or phone calls, the consent form was read aloud, and verbal consent was obtained and audio recorded. Strict confidentiality was maintained throughout the study, with participants identified only by assigned numbers. Additionally, audio-recorded interview transcripts were securely stored and will be disposed of in accordance with the university’s data protection protocol upon study completion.

### 2.8. Rigor

Several measures were implemented to enhance credibility, transferability, reflexivity, and transparency, thereby strengthening the study’s trustworthiness [34]. Credibility was ensured by recruiting individuals with firsthand experiences of prostate cancer, allowing them to share authentic insights. Transferability was enhanced through detailed descriptions of the participants, study setting, and procedures, as well as the inclusion of direct participant quotes to highlight their perspectives and comparisons with global research. Reflexivity involved critical self-reflection by the researchers on their role and potential influence throughout the study. To promote transparency and reflexivity, the interview guide, data collection methods, and data analysis procedures were thoroughly discussed among the research team to ensure rigor and consistency.

## 3. Results

### 3.1. Participants Characteristics

A total of 11 participants took part in the study, with 2 being black Irish and 9 being white Irish. The youngest participant was 62 years old, while the oldest was 81 years old, with a mean age of 72.8 years. The marital status of the participants was as follows: single (n = 3), divorced (n = 2), married (n = 6). Regarding their stage of diagnosis: three were early stage, five were intermediate stage, and three advanced or recurrent stage. Some of the participants (n = 5) were from urban cities, while the remaining (n = 6) were from rural areas. During the study four participants were working, two on the verge of retirement, and five were fully retired and not working.

### 3.2. Themes Identified

The experience of Irish men with prostate cancer revealed five major themes: (1) systemic obstacle in timely cancer detection; (2) the role of efficient system in cancer care; (3) emotional resilience in cancer recovery; (4) redefining normalcy post-treatment; (5) harnessing specialised support network in coping strategies.

### 3.3. Theme 1: Systemic Obstacle in Timely Cancer Detection

During the interviews, participants frequently expressed frustration over delays in diagnosis caused by systemic inefficiencies. Many reported long waiting times for biopsies, diagnostic imaging, and cancer referrals.

“I rang (named hospital), and I said, would you ever ring (named other hospital) and tell them to hurry up? …. and they said, oh we cannot …… it is between me and (named the other hospital)”.(p5)

Other concerns raised included a lack of awareness of alternative diagnostic centres and poor communication between healthcare providers.

“Nine months to get a biopsy ……. they should have noticed something sooner... they could have told me the biopsy can be done in a private place for a few hundred euros”.(p5)

Delays heightened stress and may cause poorer outcomes.

“The waiting list was so long; …… it felt like forever before I got a diagnosis”(p6)

“I had symptoms for months, but getting a referral took ages”(p5)

### 3.4. Theme 2: The Role of the Efficient System in Cancer Care

Many participants expressed satisfaction and gratitude for the treatment they received once inside the healthcare system. However, they also highlighted gaps in guidance on long-term management, side effects, and post-cancer care.

“The doctors and nurses were excellent …. but I had to figure out post treatment care on my own”.(p3)

Participants emphasized the need for multidisciplinary treatment and advocated for early involvement of specialists such as radiotherapists, oncologists, and other support services.

“There needs to be more multidisciplinary involvement (urologists, oncologists, radiologists, pathologists, specialist nurses, psychologists, physiotherapists, primary care physician) from the start”(p4)

This theme highlights the importance of structured follow-up procedures and patient education to improve long-term outcomes.

### 3.5. Theme 3: Emotional Resilience in Cancer Recovery

Upon receiving their diagnoses, many participants described experiencing a rollercoaster of emotions, including shock, disbelief, and fear, particularly among those with limited prior knowledge of prostate cancer.

“I did not expect it at all……. there is a lot of fear that hits all at once”(p6)

Concerns about recurrence and uncertainty also emerged, especially among those diagnosed with intermediate or advanced-stage prostate cancer.

“How long will I live”.(p2)

However, some participants demonstrated emotional adaptability, finding ways to accept their diagnosis and focus on recovery.

“Eventually I found acceptance by focusing on the present and what I can control”(p5)

### 3.6. Theme 4: Redefining Normalcy Post Treatment

The study revealed that the physical toll of prostate cancer treatment, including surgery, chemotherapy, radiotherapy, and hormonal therapy presented significant challenges such as fatigue, incontinence and erectile dysfunction.

“My body just did not feel like my own for a long time”(p6)

Participants described undergoing major adjustments that altered their activity levels, expectations, and overall sense of normalcy.

“I have adjusted to a slower pace……things are just different now”(p5)

As a result, many participants emphasized the need to redefine their sense of normalcy and adapt to new physical and emotional realities post-treatment.

### 3.7. Theme 5: Harnessing Specialised Support Network in Coping Strategies

This theme highlights the crucial role of support systems in the lives of individuals with prostate cancer. Most participants emphasized that peer support groups, family, and partner support were instrumental in helping them navigate both the emotional and physical challenges associated with the condition.

“I would not have coped as well without her (wife) constant encouragement”(p1)

“My old army buddies have been a pillar of strength through this”(p3)

Participants also discussed various support networks that addressed their specific needs.

“Connecting with others with similar situations gave me hope and practical advice”(p6)

”Attending veterans support group gave me a sense of belonging”(p6)

## 4. Discussion

This qualitative descriptive study provides a comprehensive exploration of the lived experiences of men with prostate cancer in Irish society, uncovering five key themes that encapsulate the emotional, systemic, and social dimensions of their journey and survivorship. These themes highlight the interplay between the healthcare system, personal resilience, and support networks, offering valuable insights into areas that require increased awareness and intervention. The findings revealed systemic obstacles in timely cancer detection, the role of an efficient healthcare system in cancer care, emotional resilience in cancer recovery, redefining normalcy post-treatment, and harnessing specialised support networks in coping strategies. These themes align with the Biopsychosocial Model [35], which recognizes prostate cancer survivorship as shaped by physical health challenges, emotional resilience, and social support structures.

Participants consistently reported delays in diagnosis as a significant challenge, citing prolonged waiting times for biopsies, diagnostic imaging, and hospital referrals. Early cancer detection is crucial for improving prognosis and survival rates, yet these findings align with recent studies showing that men with prostate cancer often experience longer diagnostic and treatment waiting times compared to other cancers, leading to psychological distress and potentially worse outcomes. Diagnostic delays and lack of support networks negatively impact the clinical trajectory and psychosocial wellbeing of men [36]. Fragmented communication between care services and lack of support networks contribute to cancer progression and reduced survival. These results are consistent with with studies by Mottet et al. and the American Cancer society which underscore the critical importance of early detection in improving prostate cancer outcomes globally [36,37]. However, systemic barriers such as limited healthcare access and cultural stigma often hinder early detection efforts. Additionally, delayed diagnosis increases psychological distress, with many patients experiencing uncertainty and anxiety about the disease’s progression [38]. To address these diagnostic delays and improve survival rates, streamlined referral processes, improved public awareness, and enhanced primary care provider training are necessary [39]. The Biopsychosocial Model further emphasizes the importance of integrating social health determinants into cancer detection strategies, advocating for community-based screening programs, patient navigation services, and culturally competent healthcare delivery. Delayed diagnosis in hospitals requires policy commitment, workflow redesign, patient-centred approaches, and technology integration for timely prostate cancer diagnosis, enhancing clinical efficiency and humanized care [38,39].

While 9 out of 11 of the participants expressed satisfaction and gratitude for the care they received after diagnosis, gaps in post-treatment guidance and follow-up care were evident. Effective cancer care systems require seamless coordination across diagnosis, treatment, and survivorship. However, fragmented healthcare services often result in delays, miscommunication, and suboptimal treatment outcomes [40]. These findings align with research by Garcia-Baquero et al. and Shore et al. which indicate that a well-planned, comprehensive multidisciplinary approach significantly improves treatment outcomes and patient experiences [41,42]. Participants advocated for earlier involvement of oncologists, radiotherapists, and mental health professionals, reinforcing the importance of integrated care pathways. Studies by Kord et al., Østergaard et al. and de Resende Izidoro et al. emphasize the benefits of personalized survivorship programs that address psychological, physical, and sexual health post-treatment [43,44,45]. Additionally, patient navigation initiatives which guide patients through the treatment process, have been shown to improve treatment adherence and reduce healthcare disparities [46]. The Biopsychosocial Model underscores the necessity of a patient-centred approach, integrating biological, psychological, and social factors through a multidisciplinary team. Recent studies highlight the value of biopsychosocial screening initiatives in oncology, as they help identify distress, unmet psychological needs, and barriers to treatment at early stages [47]. Moreover, ref. [48] suggests that AI-driven decision support systems and electronic medical records can enhance coordination among multidisciplinary teams and improve continuity of care.

Participants described experiencing a range of emotions upon diagnosis, including shock, fear, anxiety, and eventual acceptance. Cancer diagnosis and treatment place significant psychological burdens on patients, affecting their mental well-being and resilience. This finding aligns with studies emphasizing the role of adaptive coping strategies in managing the psychological impact of prostate cancer [49]. Emotional resilience is a crucial factor in successful recovery, requiring support systems, time, and coping mechanisms. Activities such as exercise, outdoor engagement, meditation, and creative outlets help manage stress and emotional challenges [50,51]. The Psychological Model of Coping supports these findings, highlighting adaptive coping strategies as essential in managing chronic disease stressors [52]. Professional counselling and peer support have been identified as effective in reducing depression and anxiety [53,54]. However, stigma and societal expectations often discourage men from expressing psychological distress, necessitating gender-sensitive interventions to improve mental health engagement [55]. Interventions such as cognitive-behavioural therapy (CBT) and mindfulness-based stress reduction (MBSR) have proven effective in reducing anxiety, depression, and fear of recurrence, leading to improved quality of life and treatment adherence [51,56]. The Biopsychosocial Model suggests that incorporating self-efficacy strategies, goal setting, and meaning-making interventions into cancer care can further support resilience-building.

A total of 10 out of 11 participants faced significant physical and emotional challenges of post-treatment, including fatigue, urinary incontinence, and erectile dysfunction, which often disrupted their sense of identity and masculinity. Studies by Chung et al. and Talvitie et al. corroborate these findings, demonstrating that prostate cancer treatment frequently impacts masculinity and self-perception [57,58]. To navigate these changes, participants recalibrated their expectations, developed new routines, and adopted coping mechanisms. Research suggests that sexual health awareness programs, physical rehabilitation, and fatigue management initiatives significantly improve post-treatment quality of life [59]. The Biopsychosocial Model emphasizes the importance of personalized survivorship care plans to manage long-term effects of cancer [60].

While peer support groups, mental health organizations, and advocacy groups emerged as valuable resources, 7 out of 11 participants expressed reluctance to seek help. Studies by Wilkerson et al. indicate that older men often avoid disclosing struggles due to societal norms equating vulnerability with weakness, leading to increased depression, anxiety, and social isolation [61]. The Biopsychosocial Model highlights the need for comprehensive support networks beyond clinical settings. Ref. [62] found that online support communities, peer-led groups, and men’s advocacy organizations help cancer patients express emotions, share experiences, and build community connections. Expanding access to support networks in rural and underserved areas is essential to addressing disparities in prostate cancer care. Refs. [63,64] emphasize that patient-centred support groups foster resilience, motivation, and improved quality of life.

The high prevalence of prostate cancer in Ireland is multifactorial, influenced by demographic trends, genetic predisposition, lifestyle factors, and robust cancer surveillance systems. In Ireland the primary risk factor for prostate cancer like many other countries is the aging population. Prostate cancer is the most diagnosed cancer in Irish men, with most cases occurring in those over 60 years of age [65]. Genetic and familial risk factors play a significant role and men with a family history of prostate cancer are at higher risk, and certain inherited gene mutations (e.g., BRCA1 and BRCA2) can increase susceptibility and Irish men have a relatively high burden of hereditary cancer risk, which may contribute to incidence [66]. Lifestyle and dietary factors may also influence prevalence. Diets high in animal fats and red meat, which are common in Western countries, including Ireland, have been associated with increased risk, although evidence remains inconclusive. Another important factor is increased awareness and widespread PSA testing, especially in past decades, which has led to higher detection rates. While PSA screening has been debated for its risks of overdiagnosis, it has undoubtedly contributed to identifying more cases, including those that may have remained undetected otherwise. Lastly, population-based cancer registration in Ireland is relatively comprehensive, contributing to more accurate and complete incidence reporting compared to some other countries.

Improving delayed diagnosis in hospitals, particularly for conditions like prostate cancer, requires a multifaceted approach focused on system-level changes, better clinical practices, and increased patient engagement. Adopting structured diagnostic pathways can significantly reduce the time to diagnosis. For instance, the Rapid Assessment for Prostate Imaging and Diagnosis (RAPID) pathway demonstrated a reduction in median diagnostic time from 32.1 days to 15.9 days [67]. The integration of mpMRI into the diagnostic process has improved the accuracy of prostate cancer detection as mpMRI allows for better visualization of prostate lesions, facilitating targeted biopsies and reducing the detection of indolent tumors [68]. The formation of dedicated prostate cancer units comprising urologists, radiologists, oncologists, pathologists, and other specialists ensures comprehensive patient evaluation with regular multidisciplinary meetings to facilitate coordinated care, leading to reduced diagnostic delays and improved patient outcomes. The use of clinical decision support systems within electronic health records can prompt clinicians to follow evidence-based guidelines, ensuring timely follow-ups on abnormal findings, reduce diagnostic errors and enhance the efficiency of the diagnostic process. Implementing national guidelines, such as the NHS England’s timed diagnostic pathway, sets standards for achieving diagnosis within 28 days of referral providing a framework for consistent and timely prostate cancer diagnosis across healthcare settings [69].

### Strength and Limitation

A key strength of this study is its qualitative descriptive design, which facilitates a comprehensive understanding of patient experiences by capturing rich, detailed narratives. This approach is particularly valuable in healthcare research, as it remains closely aligned with participants’ accounts, thereby enhancing the validity and authenticity of the findings [70]. Additionally, the diversity of participants in terms of age, sexuality, and place of residence strengthens the study’s robustness, offering a broad spectrum of lived experiences. Such heterogeneity enriches the data, providing a nuanced understanding of the prostate cancer journey within the Irish context. However, some limitations must be acknowledged. The inherent nature of qualitative research, with its emphasis on in-depth exploration within specific contexts, may limit the generalizability of findings to broader populations [71]. Additionally, while the sample size was appropriate for qualitative inquiry, it may not fully capture the entire range of experiences of all men with prostate cancer in Ireland. It is important to note that in qualitative research, sample size is determined by data saturation, the point at which no new information emerges. Therefore, while the sample may appear limited, it is considered sufficient when saturation is achieved [72]. While this study provides valuable insights into the experiences of men with prostate cancer in Ireland, its qualitative nature and sample size should be considered when interpreting findings. These factors may influence the extent to which results can be generalized, but they do not diminish the depth and significance of the study’s contributions to understanding patient experiences. This was a qualitative study with voluntary participation, which inherently introduces certain sampling limitations as participants were self-selecting and may reflect groups more actively engaged with their healthcare or who have lived experience relevant to the study focus, in this case, older adults with prostate cancer. As such, while the study offers rich insights into the experiences of older men, it does not aim to generalize across all age groups and future research could benefit from including younger men diagnosed with prostate cancer to explore whether their experiences, particularly regarding diagnosis, psychological impact, and support needs, differ from those of older individuals.

## 5. Conclusions

This study contributes to the existing literature on prostate cancer by highlighting the multifaceted physical, psychological, emotional, and social factors that shape patients’ experiences. Addressing systemic disparities, enhancing peer and social support networks, and promoting multidisciplinary care are crucial steps toward fostering a more patient-centered approach in prostate cancer management. Implementing comprehensive, evidence-based survivorship care frameworks can significantly improve the quality of life for prostate cancer survivors. Future research should explore the long-term effects of integrated care models and develop targeted strategies to optimize survivorship outcomes and enhance patient well-being.

## Figures and Tables

**Figure 1 healthcare-13-01049-f001:**
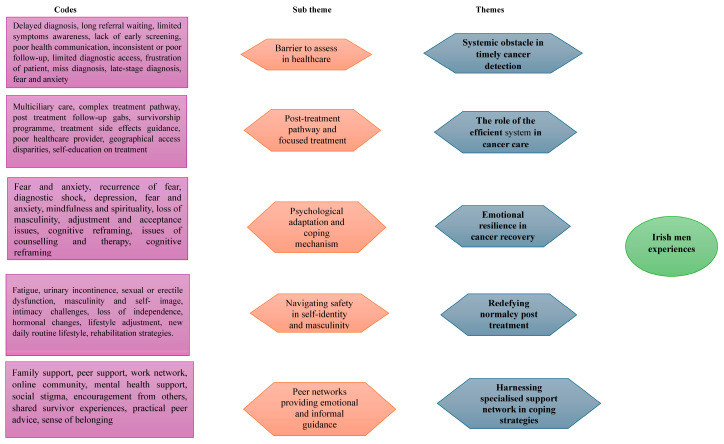
Coding and themes development.

**Table 1 healthcare-13-01049-t001:** Braune and Clarke 2021 reflective thematic analysis framework.

Phase	Description	Application to Study
Phase One: data familiarization	This stage is reading and rereading transcript to dote down initial impression	Researchers went through all 11 interviews and noted down the need information e.g., Emotional needs, unmet needs.
Phase Two: Code generation	At this stage, one systematically looks for meaningful features or patterns across all data inductively and reflectively.	Researchers generated codes like shock, fear, anxiety with the help of a qualitative analysis software.
Phase Three: Searching of theme	Codes are grouped together to make meaningful sub-themes or themes.	Generated codes were used by researchers to form subthemes like psychological adaptation and coping mechanisms.
Phase Four: Reviewing of themes generated	Generated themes reviewed in relation coded data ensuring correlation, cohesion and accuracy.	Sub-themes reviewed again in relation to the codes to help ensure correlation and distinctions.
Phase Five: Naming and Defining themes	Finally theme clearly defined, named and described.	Researchers ensure identified theme final theme is distinct and has never been used or identified in other literatures.
Phase Six: Report production	Writing a narrative about the generated themes with supporting evidence.	Developed themes were used by the researchers to write a cohesive narration to make a meaningful statement.

## Data Availability

Original data will be shared upon reasonable request to corresponding author.

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
