# Peer review of "Lived Experience of Men with Prostate Cancer in Ireland: A Qualitative Descriptive Study"

_healthcare, 2025, doi:10.3390/healthcare13091049_

Round 1
Reviewer 1 Report
Comments and Suggestions for Authors
This study provides valuable insights into prostate cancer survivorship in Ireland, with potential implications for both policy and clinical practice. It follows a clear structure, from introduction to methods, results, and partial discussion, offering an in-depth exploration of systemic, emotional, and social aspects of prostate cancer care. The study’s strengths include thematic rigor, diverse participant perspectives (ages 62–81, n=11), and the use of direct quotations that deepen the understanding of key themes, such as the impactful quote, “Nine months to get a biopsy…” (p5), which highlights systemic delays in diagnosis.
Detailed Review:
The paper adheres to research conventions, ensuring a logical narrative flow that aids in reader comprehension and effectively communicates the key themes. Participant characteristics and direct quotes vividly illustrate the challenges faced by men with prostate cancer, enhancing the relatability of the findings. For example, the quote “Nine months to get a biopsy…” (p5) underscores the delays in diagnosis.
Suggestions for Improvement:
-
Preview Key Implications: Consider discussing how diagnostic delays might impact survival and the role of support networks in recovery.
-
Tighten Language: Simplify phrasing for clarity, such as changing "These delays increased stress and anxiety, potentially worsening outcomes" to "Delays heightened stress and may worsen outcomes."
-
Enhance Methodological Details: Include context on participant recruitment, such as urban or rural clinic locations, and specify the time since diagnosis (e.g., "Participants had been diagnosed between 1–5 years ago").
-
Provide Analytical Commentary: Pair quotes with brief analyses, such as interpreting the quote “We can’t” (p5) as reflecting inter-hospital coordination failures.
Section-Specific Comments:
-
Introduction: Strong global and Irish context. Consider citing a source for the statistic “1 in 6 men” (e.g., "According to the Irish Cancer Society, 1 in 6 men…").
-
Recruitment Context: While concise, the section could benefit from more demographic details, such as participants’ urban/rural locations and socioeconomic status.
-
Multidisciplinary Involvement: Clarify what “multidisciplinary involvement” means by specifying roles such as oncologists, counselors, and other healthcare professionals.
By refining these areas, the manuscript will better highlight the study's implications and improve scholarly rigor.
Comments on the Quality of English Language- Correct Typos & Standardize Capitalization: Address typos (e.g., “Redefying” to “Redefining”) and ensure consistent capitalization (e.g., "Systemic Obstacles").
Author Response
Comments: This study provides valuable insights into prostate cancer survivorship in Ireland, with potential implications for both policy and clinical practice. It follows a clear structure, from introduction to methods, results, and partial discussion, offering an in-depth exploration of systemic, emotional, and social aspects of prostate cancer care. The study’s strengths include thematic rigor, diverse participant perspectives (ages 62–81, n=11), and the use of direct quotations that deepen the understanding of key themes, such as the impactful quote, “Nine months to get a biopsy…” (p5), which highlights systemic delays in diagnosis.
Response: Thank you for your supportive feedback.
Detailed Review:
The paper adheres to research conventions, ensuring a logical narrative flow that aids in reader comprehension and effectively communicates the key themes. Participant characteristics and direct quotes vividly illustrate the challenges faced by men with prostate cancer, enhancing the relatability of the findings. For example, the quote “Nine months to get a biopsy…” (p5) underscores the delays in diagnosis.
Suggestions for Improvement:
Comment 1. Preview Key Implications: Consider discussing how diagnostic delays might impact survival and the role of support networks in recovery.
Response 1: Thank you for the suggestion (see line 364 to 398). Improving delayed diagnosis in hospitals, particularly for conditions like prostate cancer, requires a multifaceted approach focused on system-level changes, better clinical practices, and increased patient engagement. Adopting structured diagnostic pathways can significantly reduce the time to diagnosis. For instance, the Rapid Assessment for Prostate Imaging and Diagnosis (RAPID) pathway demonstrated a reduction in median diagnostic time from 32.1 days to 15.9 days (Eldred-Evans et al. 2023).
Eldred-Evans D, Connor MJ, Bertoncelli Tanaka M, Bass E, Reddy D, Walters U, Stroman L, Espinosa E, Das R, Khosla N, Tam H, Pegers E, Qazi H, Gordon S, Winkler M, Ahmed HU. The rapid assessment for prostate imaging and diagnosis (RAPID) prostate cancer diagnostic pathway. BJU Int. 2023 Apr;131(4):461-470. https://doi.10.1111/bju.15899
The integration of mpMRI into the diagnostic process has improved the accuracy of prostate cancer detection as mpMRI allows for better visualization of prostate lesions, facilitating targeted biopsies and reducing the detection of indolent tumors (Polascik et al 2014).
Polascik TJ, Passoni NM, Villers A, Choyke PL. Modernizing the diagnostic and decision-making pathway for prostate cancer. Clin Cancer Res. 2014 Dec 15;20(24):6254-7. https://doi.10.1158/1078-0432.CCR-14-0247
The formation of dedicated prostate cancer units comprising urologists, radiologists, oncologists, pathologists, and other specialists ensures comprehensive patient evaluation with regular multidisciplinary meetings to facilitate coordinated care, leading to reduced diagnostic delays and improved patient outcomes. The use of clinical decision support systems within electronic health records can prompt clinicians to follow evidence-based guidelines, ensuring timely follow-ups on abnormal findings, reduce diagnostic errors and enhance the efficiency of the diagnostic process. Implementing national guidelines, such as the NHS England's timed diagnostic pathway, sets standards for achieving diagnosis within 28 days of referral providing a framework for consistent and timely prostate cancer diagnosis across healthcare settings (NHS 2022)
National Health Service (2022) Faster diagnostic pathways Implementing a timed prostate cancer diagnostic pathway Guidance for local health and care systems. London: National Health Service. https://www.england.nhs.uk/wp-content/uploads/2018/04/B1348_Prostate-cancer-timed-diagnostic-pathway.pdf
Comment 2: Tighten Language: Simplify phrasing for clarity, such as changing "These delays increased stress and anxiety, potentially worsening outcomes" to "Delays heightened stress and may worsen outcomes."
Response 2: We appreciate your suggestion. The corrections have been done see (check line 213).
Comment 3: Enhance Methodological Details: Include context on participant recruitment, such as urban or rural clinic locations, and specify the time since diagnosis (e.g., "Participants had been diagnosed between 1–5 years ago").
Response 3: We appreciate the insight. Please check (check line 196)
Comment 4: Provide Analytical Commentary: Pair quotes with brief analyses, such as interpreting the quote “We can’t” (p5) as reflecting inter-hospital coordination failures.
Response 4: All sentences have been written in full
Section-Specific Comments:
Comment 5: Introduction: Strong global and Irish context. Consider citing a source for the statistic “1 in 6 men” (e.g., "According to the Irish Cancer Society, 1 in 6 men…").
Response 5: We appreciate the corrections. Please check line 40
Comment 6: Recruitment Context: While concise, the section could benefit from more demographic details, such as participants’ urban/rural locations and socioeconomic status.
Response 6: Thank you for the correction. Please check 196 to 203
Comment 7: Multidisciplinary Involvement: Clarify what “multidisciplinary involvement” means by specifying roles such as oncologists, counselors, and other healthcare professionals.
Response 7: thank you for the suggestion. Explained in brackets line 232 - 233

Reviewer 2 Report
Comments and Suggestions for Authors
- The authors have used sampling size of 11, can the authors provide a control group for the study?
- Can the authors provide the data analysis using Braun and Clark's approach in a tabular form?
- In the study, age group between 62 to 81 is included, what about the individuals that are middle aged or young, is prostate cancer more common above the age of 60 in Ireland?
- Why prostate cancer is prevalent in Irish population?
- How delayed diagnosis can be improved in hospitals?
Author Response
Comment 1: The authors have used sampling size of 11, can the authors provide a control group for the study?
Response 2: We appreciate your suggestion as this is a qualitative study no control group was used or would it be appropriate
Comment 2: Can the authors provide the data analysis using Braun and Clark's approach in a tabular form?
Response 2: Thank you for pointing out this. Table added for clarity (see line 165)
Comment 3: In the study, age group between 62 to 81 is included, what about the individuals that are middle aged or young, is prostate cancer more common above the age of 60 in Ireland?
Response 3: We appreciate your suggestion. This was a qualitative study with voluntary participation in addition according to the National Cancer Registry Ireland (NCRI, 2022), the median age at diagnosis for prostate cancer in Ireland is approximately 68 years, with most cases occurring in men over the age of 60. We acknowledge sample as a limitation in the study.
Comment 4: Why prostate cancer is prevalent in Irish population?
Response 4: Thank you for the suggestion (see line 364 to 398). The high prevalence of prostate cancer in Ireland is multifactorial, influenced by demographic trends, genetic predisposition, lifestyle factors, and robust cancer surveillance systems. We have added to the discussion – In Ireland the primary risk factor for prostate cancer like many other countries is the aging population. Prostate cancer is the most diagnosed cancer in Irish men, with the majority of cases occurring in those over 60 years of age (NCRI, 2022). Genetic and familial risk factors play a significant role and men with a family history of prostate cancer are at higher risk, and certain inherited gene mutations (e.g., BRCA1 and BRCA2) can increase susceptibility and Irish men have a relatively high burden of hereditary cancer risk, which may contribute to incidence (Hsieh et al. 2024).
Hsieh, AR., Luo, YL., Bao, BY. et al. Comparative analysis of genetic risk scores for predicting biochemical recurrence in prostate cancer patients after radical prostatectomy. BMC Urol 24, 136 (2024). https://doi.org/10.1186/s12894-024-01524-6
Lifestyle and dietary factors may also influence prevalence. Diets high in animal fats and red meat, which are common in Western countries, including Ireland, have been associated with increased risk, although evidence remains inconclusive. Another important factor is increased awareness and widespread PSA testing, especially in past decades, which has led to higher detection rates. While PSA screening has been debated for its risks of overdiagnosis, it has undoubtedly contributed to identifying more cases, including those that may have remained undetected otherwise. Lastly, population-based cancer registration in Ireland is relatively comprehensive, contributing to more accurate and complete incidence reporting compared to some other countries.
Comment 5: How delayed diagnosis can be improved in hospitals?
Response 5: Thank you for the suggestion (see line 364 to 398). Improving delayed diagnosis in hospitals, particularly for conditions like prostate cancer, requires a multifaceted approach focused on system-level changes, better clinical practices, and increased patient engagement. Adopting structured diagnostic pathways can significantly reduce the time to diagnosis. For instance, the Rapid Assessment for Prostate Imaging and Diagnosis (RAPID) pathway demonstrated a reduction in median diagnostic time from 32.1 days to 15.9 days (Eldred-Evans et al. 2023).
Eldred-Evans D, Connor MJ, Bertoncelli Tanaka M, Bass E, Reddy D, Walters U, Stroman L, Espinosa E, Das R, Khosla N, Tam H, Pegers E, Qazi H, Gordon S, Winkler M, Ahmed HU. The rapid assessment for prostate imaging and diagnosis (RAPID) prostate cancer diagnostic pathway. BJU Int. 2023 Apr;131(4):461-470. https://doi.10.1111/bju.15899
The integration of mpMRI into the diagnostic process has improved the accuracy of prostate cancer detection as mpMRI allows for better visualization of prostate lesions, facilitating targeted biopsies and reducing the detection of indolent tumors (Polascik et al 2014).
Polascik TJ, Passoni NM, Villers A, Choyke PL. Modernizing the diagnostic and decision-making pathway for prostate cancer. Clin Cancer Res. 2014 Dec 15;20(24):6254-7. https://doi.10.1158/1078-0432.CCR-14-0247
The formation of dedicated prostate cancer units comprising urologists, radiologists, oncologists, pathologists, and other specialists ensures comprehensive patient evaluation with regular multidisciplinary meetings to facilitate coordinated care, leading to reduced diagnostic delays and improved patient outcomes.​ The use of clinical decision support systems within electronic health records can prompt clinicians to follow evidence-based guidelines, ensuring timely follow-ups on abnormal findings, reduce diagnostic errors and enhance the efficiency of the diagnostic process. Implementing national guidelines, such as the NHS England's timed diagnostic pathway, sets standards for achieving diagnosis within 28 days of referral providing a framework for consistent and timely prostate cancer diagnosis across healthcare settings (NHS 2022)
National Health Service (2022) Faster diagnostic pathways Implementing a timed prostate cancer diagnostic pathway Guidance for local health and care systems. London: National Health Service. https://www.england.nhs.uk/wp-content/uploads/2018/04/B1348_Prostate-cancer-timed-diagnostic-pathway.pdf

Reviewer 3 Report
Comments and Suggestions for Authors
This is a short article explaining the experience of a small cohort of prostate cancer patients from ireland. The paper is well laid out and acts as a tool for the irish healthcare system. A few minor suggestions to improve the manuscript:
- Within patient quotations, some slang is used, it might be helpful for the reader if these quotes were elaborated on or slightly reworded such that the international community can fully understand
- Suggest adding some stats of the findings e.g. 8/11 of the patients interviewed experienced delays in the diagnosis etc
- Within the text, the way the author is discussing a citation isnt standard practice for manuscripts
- Example: These results are consistent with [36,37] which underscore the critical importance of early detection in improving prostate cancer outcomes globally.
- Change to: These results are consistent with studies by Engel and Mottet which underscore the critical importance of early detection in improving prostate cancer outcomes globally [36,37].
- There are more instances, the above is given as an example
Author Response
Comment 1: Within patient quotations, some slang is used, it might be helpful for the reader if these quotes were elaborated on or slightly reworded such that the international community can fully understand
Response 1: We appreciate your suggestions, all the slangs in the results have been corrected as suggested. See line 299
Comment 2: Suggest adding some stats of the findings e.g. 8/11 of the patients interviewed experienced delays in the diagnosis etc
Response 2: Thank you for your insight, some statistics has been included as suggested. See line 302, 313, 341, and 352.
Comment 3: Example: These results are consistent with [36,37] which underscore the critical importance of early detection in improving prostate cancer outcomes globally. Change to: These results are consistent with studies by Engel and Mottet which underscore the critical importance of early detection in improving prostate cancer outcomes globally [36,37].
Response 3: We appreciate your suggestions. All such sentences have been corrected in reference [36,37], [41,42], [43,44,45], [57,58] and [61]

Round 2
Reviewer 2 Report
Comments and Suggestions for Authors
All the questions have been addressed